# Assembly and Cellular Exit of Coronaviruses: Hijacking an Unconventional Secretory Pathway from the Pre-Golgi Intermediate Compartment via the Golgi Ribbon to the Extracellular Space

**DOI:** 10.3390/cells10030503

**Published:** 2021-02-26

**Authors:** Jaakko Saraste, Kristian Prydz

**Affiliations:** 1Department of Biomedicine and Molecular Imaging Center, University of Bergen, P.O. Box 7800, 5020 Bergen, Norway; 2Department of Biosciences, University of Oslo, Boks 1072 Blindern, NO-0316 Oslo, Norway; kristian.prydz@ibv.uio.no

**Keywords:** coronavirus (CoV), virus assembly, virus egress, ER-Golgi intermediate compartment (IC or ERGIC), vesicular tubular cluster (VTC), recycling endosome (RE), Golgi ribbon, unconventional secretion, Golgi bypass, Rab1, Rab11, megavesicles

## Abstract

Coronaviruses (CoVs) assemble by budding into the lumen of the intermediate compartment (IC) at the endoplasmic reticulum (ER)-Golgi interface. However, why CoVs have chosen the IC as their intracellular site of assembly and how progeny viruses are delivered from this compartment to the extracellular space has remained unclear. Here we address these enigmatic late events of the CoV life cycle in light of recently described properties of the IC. Of particular interest are the emerging spatial and functional connections between IC elements and recycling endosomes (REs), defined by the GTPases Rab1 and Rab11, respectively. The establishment of IC-RE links at the cell periphery, around the centrosome and evidently also at the noncompact zones of the Golgi ribbon indicates that—besides traditional ER-Golgi communication—the IC also promotes a secretory process that bypasses the Golgi stacks, but involves its direct connection with the endocytic recycling system. The initial confinement of CoVs to the lumen of IC-derived large transport carriers and their preferential absence from Golgi stacks is consistent with the idea that they exit cells following such an unconventional route. In fact, CoVs may share this pathway with other intracellularly budding viruses, lipoproteins, procollagen, and/or protein aggregates experimentally introduced into the IC lumen.

## 1. Introduction

Enveloped animal viruses have been frequently used as models to study the secretory process, since the viral membrane proteins hijacking the host cell’s transport machineries follow the same intracellular itineraries as their endogenous cellular counterparts. Accordingly, after their synthesis in the endoplasmic reticulum (ER, see Appendix A), membrane glycoproteins of most enveloped viruses, such as influenza virus, move along the constitutive secretory pathway via the Golgi apparatus to the plasma membrane (PM), where the assembly of progeny viruses by budding releases them directly to the extracellular space. However, for reasons that are not completely understood certain viruses bud into the lumen of intracellular organelles, such as the ER, intermediate compartment (IC, see Appendix A), or the Golgi apparatus. This requires that the newly made virus particles enclosed in transport carriers must find their way from the early secretory route to the cell surface to be released by exocytosis. It has been generally accepted that the delivery of thes e viruses from their sites of formation to the cell exterior is also based on constitutive secretion involving passage of cargo across the cisternal Golgi stacks [1,2,3]. However, as infection of cells by these viruses is typically accompanied by major changes in Golgi organization, it has been difficult to understand why they would have developed the ability to disrupt the very organelle that they depend on for their assembly and/or cellular exit [4]. In an attempt to solve this puzzle, we address here the late replication strategies of coronaviruses (CoVs), a large family of intracellularly budding viruses. We propose that following their assembly at the IC membranes these viruses embark on an unconventional journey that leads from the early secretory pathway directly to the endosomal recycling system, explaining why their release does not depend on Golgi integrity.

## 2. The CoV Budding Compartment 

Assembly by budding into the IC lumen is a general hallmark of CoVs. Originally, electron microscopic (EM) studies of cells infected with mouse hepatitis virus (MHV) demonstrated the intracellular assembly mode of these viruses. It was initially concluded that CoV budding takes place in association with the major organelles of the secretory pathway, the ER and/or the Golgi apparatus, although virus particles were also seen in smooth-walled vacuoles of unknown character. Notably, no budding was observed at the PM [5]. To determine the precise intracellular site of virus formation, Tooze and coworkers carried out a detailed EM investigation of MHV-infected mouse fibroblasts, revealing that at early times of infection, the budding of progeny viruses begins at a pleomorphic tubulo-vesicular compartment at the ER-Golgi interface [6]. These smooth-membraned structures, apparently related to the transitional ER elements described in other cell types [7], were morphologically distinct from the rough ER and Golgi *cisternae*, although frequently found in their vicinity [6]. Moreover, initiation of O-glycosylation of the MHV M-protein—by addition of N-acetyl-galactosamine (GalNAc)—was suggested to take place at this location [8]. During the early stages of virus infection, the “budding compartment” was the only site of MHV assembly, whereas at later time points budding into rough ER was also observed [6]. Interestingly, in contrast to what was observed in mouse fibroblasts, virus formation in AtT20 pituitary tumor cells was found to be largely restricted to the stacked Golgi *cisternae*, typically occurring at their dilated rims [9,10]. 

Subsequent studies showed that the pre-Golgi compartment where CoV budding takes place corresponds to the IC, which had been introduced as a novel sorting station in bidirectional ER-Golgi trafficking [11,12,13]. Accordingly, the CoV budding compartment was shown to contain p58/ERGIC-53 (See Appendix A) [14,15,16], a cargo receptor which cycles between the ER and IC/*cis*-Golgi [13,17]. Notably, it has been reported that p58/ERGIC-53 is incorporated into forming CoV particles and might even be required for their infectivity [18]. Besides p58/ERGIC-53, the budding compartment harbors the GTPases Rab1 and Rab2 (See Appendix A), two transport machinery components operating in ER-Golgi trafficking [15,19]. Of the two proteins, Rab1 has been particularly well-characterized as a specific IC/*cis*-Golgi resident and a master regulator of ER-Golgi communication and Golgi organization [20,21,22,23]. Interestingly, the four CoVs investigated by Klumperman and coworkers, belonging to different genera (α-, β-, and γ-CoVs), were all found to employ the IC as their intracellular site of formation, while none of them assembled at Golgi membranes [14]. The two more recently emerged β-CoVs causing serious disease in humans—Severe Acute Respiratory Syndrome (SARS) and Middle East Respiratory Syndrome (MERS) CoVs—have been shown to share this property [24]. 

## 3. The Assembly and Exit Strategies of CoVs Are Not Unique

Although budding at the IC is frequently brought up as one of the distinctive features of CoVs, it seems likely that their assembly or release strategies are shared by a number of viruses that bud into the lumen of endomembrane compartments. The IC has been implicated in the complex envelopment of large DNA viruses, such as herpes and vaccinia virus [25,26,27] and may also participate in the assembly of hepatitis B virus [28,29]. Regarding RNA viruses, rubella virus—a togavirus—has been found to bud either at the PM or in the Golgi area, depending on the host cell type [30]. Furthermore, it is generally accepted that bunyaviruses, such as Uukuniemi virus (UUK) and Hantaan virus (HTNV), employ the Golgi apparatus as their intracellular site of assembly [1,31,32,33,34]. Interestingly, however, immuno-EM analysis of BHK-21 cells at an early stage of UUK infection showed that, in addition to the dilated ends of the Golgi *cisternae,* virus budding also occurs at p58/ERGIC-53-containing peripheral and central IC elements [35]. Moreover, shifting cells to 15 °C, which blocks ER-to-Golgi transport at the level of the IC [11], resulted in the accumulation of the viral nucleocapsid (N) and membrane proteins at this compartment, providing further evidence for its role in UUK assembly [35]. Similarly, the N protein of HTNV is targeted to the IC, but not to the Golgi apparatus, apparently due to dynein-based transport of viral ribonucleoproteins or nucleocapsids along microtubules (MTs) [36]. Moreover, membrane association of the N protein was not affected by Brefeldin A (BFA, See Appendix A), a fungal compound, which disassembles the Golgi stacks but leaves the stable IC elements largely unaffected [13,36]. 

Flaviviruses, such as hepatitis C virus (HCV) and Zika virus, bud into the lumen of a modified ER-derived compartment—or an “ER-Golgi hybrid compartment” [37]—and are then thought to be transported along the constitutive secretory pathway via the Golgi apparatus to the extracellular space [3]. Some studies also suggest the participation of the IC in the assembly and/or release of flaviviruses. For example, the IC-associated GTPase Rab1 has been implicated in the assembly of the classical swine fever virus (CSFV) [38]. Moreover, the KDEL-receptor, which cycles at the ER-Golgi interface and predominantly localizes to the IC/*cis*-Golgi membranes [12,13], is required for the anterograde transport of Dengue virus in the early secretory pathway [39]. Finally, rotaviruses assemble by budding into the ER lumen, but then lose their membrane in a poorly understood process. Notably, their export from epithelial cells appears to involve an unconventional lipid raft-dependent pathway that bypasses the Golgi apparatus [40,41]. 

In summary, the above studies clearly indicate that the assembly and/or exit strategies of CoVs are shared by other intracellularly budding viruses. In particular, the late steps in the replication of bunyaviruses and CoVs may in fact be more similar than has previously been considered, as also indicated by the suggested incorporation of p58/ERGIC-53 into both types of viral particles [18]. However, infection of cells by bunyaviruses could result in more extensive or rapid dilation of the IC and/or Golgi membranes (see below), explaining why their budding has been thought to be restricted to the latter organelle [1,34]. Whether or not the IC plays a role in the assembly of flaviviruses is not clear, but they are nonetheless expected to pass through this compartment on their way out of the infected cells [42].

## 4. CoV Assembly at the IC Membranes

The pleomorphic CoV virions display considerable size heterogeneity, ranging from 100 to 160 nm in diameter (Figure 1A). They consist of an inner nucleocapsid and a surrounding envelope—a lipid bilayer, which is derived from the IC during the budding process and incorporates the viral membrane proteins (abbreviated S, M and E). To form the nucleocapsid multiple copies of the N protein bind in a beads-on-a-string fashion to the single-stranded genomic RNA. The ensuing ribonucleoprotein folds upon itself, forming the helical nucleocapsid [43,44] (Figure 1A). The spike protein (S) is a large trimeric glycoprotein containing N-linked glycans. It protrudes from the envelope giving the virion its corona-like appearance and plays key roles during virus attachment and entry into the host cells [45,46]. The triple-spanning membrane (M) protein is the major protein component of the viral envelope. By undergoing homo-oligomerization and interacting with the other structural proteins (N, S, and E), as well as the genomic RNA, the M protein is thought to give the virion its shape [45,46]. Depending on the CoV, M is variably modified by N-linked or O-linked sugars—sometimes by both. Finally, the envelope (E) protein is a small single-spanning integral membrane protein, which is normally not glycosylated, but modified by palmitoylation. Notably, although abundantly expressed in the infected cells, E is present only in small numbers in the virus particles. Nevertheless, studies of the formation of virus-like particles (VLPs) revealed an essential role of E in CoV assembly [47]. By interacting with M protein, the E protein could, for example, induce membrane curvature during budding, or act as a scission protein to complete virus assembly [4,48]. Additionally, the multifunctional E protein homo-oligomerizes to form a pentameric ion channel (viroporin) and plays an essential role during the release of progeny viruses [4,48].

In analogy to viruses budding at different PM domains of epithelial cells [49], the intracellular mode of assembly of CoVs requires that the topologically distinct virus components—integral membrane proteins (M, E and S) and cytoplasmic nucleocapsids—efficiently meet at the same cellular location, in this case at the IC membranes (Figure 1B). Following their synthesis in the ER the viral membrane proteins are incorporated into transport vesicles at ER exit sites (ERES) and delivered to the IC in a process that is quite well known. By contrast, it remains more enigmatic where and how the N proteins interact with the newly made RNA genomes to assemble viral nucleocapsids, and how the latter are specifically targeted to the site of budding. Interestingly, however, recent studies have revealed that the interaction of N proteins with the genomic RNA drives their assembly into phase separated condensates, a process which is most likely linked to CoV budding [50,51]. Furthermore, it is tempting to speculate that the delivery of CoV nucleocapsids to the IC shares similarity with the suggested motor-dependent transport of the bunyavirus N protein along MTs [36]. When a sufficiently high local concentration of viral structural components is achieved, virus budding into the IC lumen could be largely driven by their specific interactions (Figure 1B). The most abundant membrane protein M may nucleate virus assembly due to its interactions with the genomic RNA and the other structural proteins [52,53,54]. The cytoplasmic domains of M proteins homo-oligomerize, possibly building a special matrix-type structure at the inner surface of the envelope [43]. Notably, in contrast to most membrane viruses, the nucleocapsids are not necessary for the budding of certain CoVs, with coexpression of the M and E proteins providing a minimalistic system for VLP formation [47,55].

The first clue to the mechanisms that determine the site of CoV budding came from studies on the multispanning M protein, the master protein of virus assembly. Mutational analysis of the individually expressed M protein of avian infectious bronchitis virus (IBV) revealed motifs in its first transmembrane segment that dictate its retention to the IC/*cis*-Golgi region [56,57]. However, subsequent studies showed that, in contrast to the IBV M protein, the corresponding proteins of two other CoVs—MHV and MERS-CoV—can reach the *trans*-Golgi/TGN [14,58,59]. Oligomerization of the M protein also contributes to its retention [54,60]. The other CoV membrane proteins—S and E—are also largely retained intracellularly, accumulating in the perinuclear IC/Golgi area. The IBV S protein contains a di-lysine ER-retrieval signal in its cytoplasmic tail, which by interacting with IC/*cis*-Golgi localized COPI (See Appendix A)-coats contributes to its concentration near the site of virus assembly [61]. This motif is also found in the S proteins of other CoVs [54,62,63] and may ensure their efficient coalescence with the other viral membrane proteins [62,64]. Similar to S, the E protein is not specifically targeted to the IC membranes but displays a broader distribution in the Golgi region [4,65,66]. A recent study demonstrated the localization and mobility of the MHV E protein in the IC and early Golgi membranes [67]. 

The observed differential localization of the M protein illustrates that the retention of CoV envelope proteins does not suffice to determine the site of budding, although it is most likely a major requirement [53]; therefore, there must be other contributing factors. While glycosylation of the viral membrane proteins is dispensable for CoV assembly, other post-translational modifications of these proteins—such as palmitoylation of the E protein—may play a role [48]. The nonstructural and accessory proteins (ORFs) encoded by CoVs may also contribute to virus assembly, for example, by interacting with cargo receptors or other transport machinery, or by affecting lipid biosynthesis [53,68,69]. Indeed, the pleomorphic nature of the virions may suggest the participation of host proteins in CoV formation. Moreover, specific lipid composition of the IC membranes is expected to play an important role in the budding event [70]. Regarding the role of the cytoskeleton, the M protein has been shown to interact with actin filaments, which could provide a force-generating system for the assembly process [71]. 

Interestingly, studies carried out with IBV have provided evidence for the existence of two functional pools of the E protein in CoV-infected cells. An oligomeric pool participates in virus assembly, while a monomeric pool—evidently by interacting with a host component(s)—neutralizes the luminal acidic pH of secretory compartments. Paradoxically, the activity of this monomeric pool not only supports virus release but also induces Golgi disassembly [72,73,74] (see below). Since the acidic luminal pH of the IC [75,76,77] is most likely also affected by the E protein ion channel, it is evidently not one of the determinants of CoV budding. Accordingly, the carboxylic ionophore monensin, which neutralizes the IC and Golgi lumen, does not block CoV assembly, but results in the pile-up of immature virus particles within the IC elements [16].

## 5. A Viral Perspective on IC Organization 

While it has been well established that CoVs preferentially employ IC as their budding site, the precise membrane subdomain of the IC where virus assembly takes place has not been determined. However, it is possible to address this question by considering some of the models presented on the structural organization of this organelle [13]. For example, the individual IC elements have been designated as vesicular tubular clusters (VTCs, See Appendix A), a morphological term referring to assemblies of small vesicles and tubules, which are encountered at the ER-Golgi boundary and contain newly synthesized secretory or PM proteins [78]. In the context of the different models on IC dynamics the VTCs have been viewed either as stationary structures, or dynamic entities displaying MT-dependent motility (see below). Notably, however, both analytical cell fractionation and EM studies have convincingly demonstrated that the IC includes an additional saccular component, up to 0.5 μm in diameter [11,13,76,79]. With the small vesicles and narrow tubules lacking the necessary luminal space to accommodate the large CoV particles, these pleomorphic saccular IC elements, with their ability to expand [8,13], provide a suitable site for virus budding (Figure 1B). As suggested by the above discussed studies with IBV, the E protein, by raising the luminal pH of secretory compartments [74], could bring about further dilation of these elements, thereby increasing their capacity to house large numbers of newly formed progeny viruses [80].

The use of fluorescent Rab1 as a reporter in living cells showed that the dynamic IC network includes two types of elements capable of performing long-distance movements; namely, highly dynamic tubules and large globular structures, which before becoming mobile may remain stationary for longer periods of time [13]. Rather than the VTCs, these globular structures most likely correspond to the saccular IC elements described above, as demonstrated by their ability to change their shape and elongate as they begin to move [81,82,83,84]. These imaging results on IC dynamics support the scenario shown in Figure 1B, suggesting that the same saccular IC domains, where CoV assembly initially takes place, could directly develop into the specialized carriers, which are required to transfer the large-sized virus particles from their sites of budding at the IC to the central Golgi region. This is in agreement with immuno-EM data on free CoV-containing vacuoles labeled with the IC marker p58/ERGIC-53 [16,80].

A long-standing model that seems to have retained much of its popularity considers the IC as a collection of dynamic ER-to-Golgi transport intermediates. As stated above, the mobile IC elements observed in living cells have even been proposed to correspond to individual VTCs [85]. This view of the IC as a transient compartment has also strongly influenced the ideas on how CoVs are transported through the Golgi system. According to one possible scenario, after moving along MT-tracks from the widespread ERES to the Golgi region, the pleomorphic IC structures could generate the central Golgi organization by transforming into *cis*-Golgi *cisternae* (Figure 2). Thus, based on the generally accepted cisternal maturation or progression models on Golgi dynamics [86], as well as numerous EM studies showing the segregation of the large-sized CoVs to the dilated ends of the Golgi *cisternae*, their passage across the Golgi stacks could simply be dictated by the movement of the *cisternae* in *cis*-to-*trans* direction (Figure 2). 

Alternatively, considering the Golgi stacks as more stationary entities, the incoming IC carriers containing CoVs would be expected to undergo fusion with the *cis*-Golgi membranes. As a result of their lateral segregation within the *cisternae*, the progeny viruses could become enclosed in “megavesicles”, which by pinching off and fusing with the dilated cisternal ends constitute a potential mechanism for the transport of large cargo across the Golgi stacks [87] (Figure 2). This “rim progression model” of Golgi dynamics [88] was originally based on experimental deposition of large protein aggregates (ca. 0.4 μm in diameter) in the lumen of IC and *cis*-Golgi elements during the incubation of cells at 15 °C [87]. Notably, EM analysis of their synchronized passage across the Golgi stacks, achieved by temperature shift experiments, showed that these aggregates—like CoVs—are preferentially found at the dilated ends of the *cisternae*, suggesting that the two types of cargo share a similar pathway.

A common denominator of these two transport mechanisms for CoVs is that they both require the structural integrity or functionality of the Golgi stacks. For example, Golgi perturbation by monensin inhibits the release of the CoV transmissible gastroenteritis virus (TGEV) [16]. Thus, although the precise route and mechanism of transport of CoVs has not been defined, the general consensus has been that they employ or somehow modify the constitutive biosynthetic-secretory pathway to gain exit from their host cells [4,45,46]. Accordingly, following their transfer across the Golgi stacks, the progeny viruses arrive at the *trans*-Golgi network (TGN), where they are thought to be sorted into specialized post-Golgi carriers for subsequent delivery to the cell surface [89] (Figure 2).

Challenging the above described “transient compartment model” of the IC, more recent studies have emphasized the permanent nature of this organelle [13]. On one hand, it has been suggested that the tubulovesicular IC elements form a stationary compartment close to ERES, which communicates with the ER and *cis*-Golgi via distinct transport carriers [90]. On the other hand, monitoring Rab1 dynamics in living cells provided strong evidence indicating that the above described saccular and tubular IC elements establish an interconnected membrane system that persists throughout the cell cycle [13,81,82,91]. While the former model would require the formation of specialized large carriers for anterograde transport of CoVs between the IC and *cis*-Golgi, the latter model retains the main thesis of the transient compartment model; namely, that the large saccular IC elements themselves act as dynamic CoV carriers.

## 6. Golgi Stack-Independent Secretion of CoVs

A number of studies have demonstrated that the IC elements defined by p58/ERGIC-53 or Rab1 are not restricted to the ER-Golgi boundary, but also present in the pericentrosomal region and at the cell periphery [81,82,91,92]. Moreover, there is increasing evidence supporting the notion that, in addition to its well-established role in ER-Golgi communication, the IC participates in Golgi-independent trafficking of both conventional (signal peptide-containing) and unconventional (leaderless) secretory proteins [82,93,94,95,96]. Strikingly, the widely distributed IC elements are spatially linked to recycling endosomes (REs, See Appendix A) defined by Rab11, and close association of these dynamic membrane networks is still maintained when Golgi stacks are disassembled using BFA, which removes membrane-bound clathrin- and COPI-coats [13,82,95]. Since cell surface delivery of many proteins is unaffected by BFA, it appears that the BFA-resistant IC-RE connections form the basis for unconventional secretory pathways that bypass the Golgi stacks [13,94,97,98]. Recently, we suggested that these permanent membrane networks also meet at the noncompact zones of the Golgi ribbon (See Appendix A), establishing “linker compartments” that act in membrane trafficking and the biogenesis of the Golgi stacks, thereby dynamically joining the stacks into a continuous ribbon [13,99]. This idea is in accordance with the intimate association of the IC elements and REs with the *cis-* and *trans*-aspects of the Golgi apparatus, respectively [13,99,100,101]. 

Based on functional connections between the IC elements and REs within the Golgi ribbon and at the cell periphery [99], we propose an alternative pathway for CoV release, as schematically shown in Figure 3. We suggest that MT-dependent movements of the CoV-containing IC carriers from peripheral ERES to the cell center direct them preferentially to the noncompact zones of the Golgi ribbon, where the virus particles are transferred directly from the IC elements to REs based on a possible maturation process (see below). An additional role of the saccular IC elements in the generation of new Golgi *cisternae* [11,13,99] could explain the presence of CoVs at the dilated rims of the Golgi stacks. Accordingly, we propose that the recycling endosomal system, rather than the TGN, is responsible for the generation of the post-Golgi carriers that deliver the CoV particles to the cell surface to be released by exocytosis. It is also possible that CoVs employ a more direct pathway from peripheral ERES to the vicinity of the PM [81,92], where an interaction of the IC elements with peripheral REs allows their delivery to the extracellular space (Figure 3).

Importantly, the transport route(s) implicated here in CoV egress (see Figure 3) are known to be resistant to BFA [81,82,94,95]. However, investigations on how BFA affects the assembly or release of CoVs (or other viruses) have been complicated by the fact that this drug may inhibit both early and late steps of virus replication [102,103]. An EM study on the morphogenesis of TGEV, focusing on early times post infection (4–8 h), showed that BFA does not affect CoV assembly in the IC, but actually increases the number of budding profiles. While maturation of the virus was shown to be affected by BFA, as evidenced by intracellular accumulation of precursor forms of TGEV virions, the effect of the drug on virus release was not directly addressed [104]. Interestingly, a recent study from Nihal Altan-Bonnet’s lab demonstrated that BFA does not interfere with the cellular exit of CoVs [105]. When the effect of BFA was determined during different times of exponential virus release, i.e., between 6 and 14 h post infection, no significant reduction in the amount of extracellular virus was seen in the drug-treated cells, as compared to the untreated controls. The authors concluded that CoV release does not depend on the classical biosynthetic-secretory pathway, but instead involves the endo-lysosomal system, with virus egress occurring from lysosome-like organelles [105].

How about other viruses budding intracellularly? Although there are conflicting reports on the effect of BFA on flavivirus release, some studies suggest that it inhibits productive infection only when added early after the start of infection, but loses its effect if added later, during maximal virus assembly and release [37,42,106]. Moreover, blocking constitutive secretion via combined depletion of the GTPases Arf1 and Arf4—which, like treatment of cells with BFA, leads to the disassembly of the COPI coats—exerts only a moderate effect on Dengue virus release, indicating that this flavivirus follows an unconventional COPI-independent secretory pathway [39].

The endosomal recycling apparatus—consisting of both peripheral REs and the pericentrosomal endocytic recycling compartment (ERC)—has been shown to support the assembly and release of many viruses [107,108]. Indeed, the localization of a subdomain of the IC next to the centrosome [82] suggests that the assembly and/or release of CoVs may involve a functional connection of this domain with the ERC (Figure 3). Notably, REs have also been implicated in Rab11- and Rab8-dependent transport of bunyaviruses from the Golgi region to the PM of epithelial cells [109]. Similarly, studies of flaviviruses—West Nile virus (WNV) and hepatitis C virus (HCV)—revealed the localization of virus particles to perinuclear REs and their Rab11- or Rab8-dependent transport to the cell surface [110,111]. Additional studies employing live imaging have supported the role of the endosomal system in unconventional Golgi-independent release of HCV [112,113].

In conclusion, the observations regarding the effect of BFA on virus egress and the role of the endosomal recycling system and Rab11 in this process collectively support our model on a Golgi bypass route for virus release (Figure 3) that—besides different CoVs—is expected to be shared by other viruses that assemble by budding into compartments of the early secretory pathway.

## 7. Mechanisms and Consequences of CoV-Induced Golgi Disassembly 

Obviously, the above presented models on virus release illustrate the early stages of CoV infection, preceding virus-induced Golgi fragmentation and disassembly of the cisternal stacks (Figure 2 and Figure 3) [9,72,80,114,115]. The gradual replacement of the stacked Golgi organization by smooth-walled vacuoles containing progeny viruses is generally considered to result from the dilation of Golgi *cisternae*. As shown by studies of IBV, this morphological change is most likely the outcome of one of the activities of the E protein, which by interacting with host protein(s) neutralizes Golgi pH [74], and thereby could for instance affect the machinery responsible for Golgi stacking. The swelling of Golgi *cisternae* could also be caused by the luminal accumulation of large-sized virus particles [116]. The proposed dual roles of the “linker compartments” at the noncompact zones of the Golgi ribbon in CoV release and biogenesis of the Golgi stacks [99] (Figure 3) introduce a third option. Accordingly, instead of directly targeting the Golgi *cisternae*, the E protein activity could bring about neutralization of the acidic IC elements and REs [75,76,77,117], resulting in their dilation. Due to the transmembrane effects of these luminal changes on components of the cytoplasmic transport machinery, the function of these compartments in cisternal biogenesis would be hampered, giving rise to the fragmentation of the Golgi ribbon and dissolution of the Golgi stacks. Compared to the earlier models (Figure 2), this third alternative scenario would better explain the fact that CoV trafficking remains unaffected by the dramatic Golgi rearrangements, which take effect before maximum virus release. Indeed, the enlargement of the original CoV carriers would facilitate efficient transport of increasing numbers of progeny viruses. Furthermore, this scenario could explain how the activity of the E protein can support the release of IBV but inhibit PM delivery of conventional secretory and membrane proteins [72]. 

However, our model on the function and reorganization of secretory endomembrane compartments in CoV-infected cells (Figure 3) raises the question of how adequate modification and processing of the viral glycoproteins—as important prerequisites for CoV maturation, infectivity and pathogenesis [16,45,89,104]—can be ensured? Namely, analysis of released virus particles has shown that, although the glycan patterns of the hyperglycosylated S protein of SARS-CoV are highly heterogeneous, the protein is subjected to considerable terminal modification by Golgi enzymes [118]. Additionally, the S protein of certain CoVs is cleaved, and partially activated, by the proprotein convertase furin, which cycles between endosomes and the *trans*-Golgi/TGN [119,120]. Since it is very likely that the “linker compartments” would continuously communicate with the Golgi stacks via tubular and vesicular trafficking [13,99], the possibility exists that their inability to form new *cisternae* triggers the redistribution of Golgi enzymes into the IC and endosomal networks. For instance, the observed terminal glycosylation and furin-mediated processing of the S protein would not provide proof for conventional Golgi passage of CoVs, but instead take place in the endosomal carriers that deliver the viruses to the cell exterior (Figure 3). The localization of *cis*- and *trans*-Golgi proteins to the large CoV-containing carriers is in accordance with this possibility, as well [80].

The above scenario is also supported by early studies by Tooze and coworkers regarding O-glycosylation of the MHV M-protein [8]. As mentioned above, their study showed that this protein gets its proximal GalNAc added in the CoV budding compartment, indicating that GalNAc-transferase 2 (GalNAcT2), the enzyme responsible for this modification, resides in the IC. Although subsequent studies with uninfected cells showed that GalNacT2 normally localizes to more distal Golgi compartments [121], they also indicated that the enzyme continuously cycles between the Golgi apparatus and the IC [122]. Thus, CoV infection—by changing the luminal conditions of the IC—most likely results in redistribution of GalNAcT2 to the IC, making it reactive for the GalNAc-specific lectin *Helix pomatia* [8,15]. Interestingly, similar observations regarding relocalization of GalNAcT2 have been made in cancer cells [123]. Therefore, although the morphological picture may vary greatly, comparable alterations to those seen in CoV-infected cells—affecting the luminal conditions of secretory compartments and consequently the intracellular distribution of Golgi components—could take place under different experimental, physiological and pathological situations [99,124,125].

## 8. Passage of Large Cargo across the Golgi Ribbon

Besides certain viruses and engineered protein aggregates, there are also physiological cargo molecules that soon after their synthesis assemble into large filament bundles or particles that must be packed into specialized carriers so that they can proceed along the secretory pathway. These include precursors of various extracellular matrix proteins, like procollagens, as well as lipoprotein particles, including chylomicrons and very low-density lipoproteins (VLDLs) [116,126,127,128]. Following their transport to the Golgi apparatus, these large cargoes are typically encountered at the dilated rims of Golgi *cisternae* [129,130,131,132,133]. Thus, considering the proposed model (Figure 3), this common morphological feature could imply that they follow a similar unconventional pathway as CoVs—and possibly also bunya- and flaviviruses—during their secretion. However, speaking against this option, detailed EM analysis of synchronized transport of procollagen type I in cultured fibroblasts lead to the conclusion that its Golgi passage is based on cisternal maturation [133].

Interestingly, however, studies of flaviviruses have yielded results supporting the general idea that intracellularly budding viruses hijack pre-existing pathways and transport mechanisms that are normally employed by physiological large cargo [116]. For example, late steps of hepatitis C virus (HCV) replication are closely linked to the biosynthesis of lipoproteins. The proper assembly and secretion of HCV particles depend on the incorporation of certain VLDL components (such as ApoB100 and ApoE) into the virus particles, facilitating their maturation and uptake into hepatocytes [3]. Moreover, the secretion of both VLDL and HCV has been reported to depend on functional Rab1 and Rab11, indicating that their intracellular itineraries include passage through the IC and REs [110,134]. Regarding the mechanisms of formation of large cargo containers, recent studies revealed that the build-up of procollagen-containing carriers at ERES involves the IC as a membrane source [135]. While the precise mechanism of “megacarrier” formation remains to be defined [136], an important component of this process, TANGO1, which tethers the IC elements at ERES, has been shown to interact with IC/*cis*-Golgi localized transport machinery, including Rab1 and its partners GM130 and GRASP65 [137]. Initially identified as a collagen receptor, TANGO1 also interacts with ApoB100 in the transport of chylomicrons and VLDLs at the ER-Golgi boundary [138,139]; however, it’s possible role in the formation of CoV-containing IC carriers remains a subject of further study. 

The actual mechanisms for Golgi passage of large cargo remain poorly understood. As discussed above, the rim progression hypothesis postulates that the “megavesicles” containing artificial protein aggregates bud from and fuse with the ends of Golgi *cisternae*, possibly employing the same transport machinery—such as COPI, Golgi tethers and SNAREs—that mediates intra-Golgi trafficking by traditional small transport vesicles [87]. The proposed fission and fusion events could take place at the rims of neighboring stacks, possibly explaining why the transfer of large cargo depends on an intact Golgi ribbon maintained by the MT cytoskeleton, and therefore is inhibited when these filaments are depolymerized by nocodazole [140]. Alternatively, intact MT tracks could be required to ensure the meeting of the incoming IC elements and REs at the noncompact zones of the Golgi ribbon and to mediate their communication to bring about the passage of large cargo at these sites [99]. 

Surprisingly, the megavesicles carrying protein aggregates were shown to establish a “fast track” across the Golgi stacks [87]. Indeed, it is difficult to appreciate how—during 10 min at 20 °C—the initially IC/*cis*-Golgi localized luminal aggregates could reach the *trans*-Golgi by being successively incorporated into a series of large transport carriers. The model in Figure 3, proposing direct IC-RE communication [13,99], could explain both the observed rapidity of transport, as well as its independence of classical protein coats; that is, its resistance to BFA. Based on studies of endosomal maturation [141], an attractive mechanism for this transport step would be a Rab cascade that regulates the transformation of the CoV-containing IC elements into endosomal post-Golgi carriers (Figure 3). Such a mechanism would be economical, since the enclosed cargos would never leave the luminal space of their original carriers. The operation of Rab cascades has previously been implicated in traditional intra-Golgi trafficking [142]. Such regulatory cascades at the noncompact zones of the Golgi ribbon, which in addition to Rab1 and Rab11 may include other Rabs, such as Rab6, remain to be defined in mammalian cells. By contrast, the coordinated functions of the yeast counterparts of Rab1 and Rab11—Ypt1 and Ypt31/32—have been well established in the secretory trafficking of *S. cerevisiae* based on cisternal maturation [143,144,145]. An obvious difficulty with this comparison has been that the proposed interacting compartments in mammalian cells [97] traditionally belong to distinct transport systems that are expected to be separated by the Golgi stacks. However, this may not be an insurmountable obstacle, since there is evidence indicating that the successive functions of Ypt1 and Ypt31/32 in the yeast secretory pathway take place at the level of early Golgi elements and a late compartment displaying combined *trans*-Golgi and endosomal characteristics [146]. 

## 9. Summary and Perspectives

Previously, we have suggested that the noncompact zones alternating with the cisternal stacks in the Golgi ribbon are occupied by pleomorphic IC elements and REs, which by operating in the biogenesis of the Golgi stacks dynamically join them into a continuous structure. We further argued that these “linker compartments” also communicate with each other, establishing a passageway for bidirectional trafficking across the Golgi ribbon that bypasses the cisternal stacks [99]. Here, we applied this Golgi model to address the pathways and mechanisms that operate in cellular egress of CoVs. Our model is in part based on observations of how the biosynthetic (IC) and endosomal networks, which generate the central linker compartments, persist and maintain their dynamic properties when the Golgi stacks are disassembled by BFA [13,82,95,147]. Moreover, the functional connections of the two membrane networks are maintained, allowing a selection of newly synthesized proteins and lipids to reach the cell surface in a Golgi-independent manner [82,94,95,97]. Therefore, the recent study showing that the cellular exit of CoVs is unaffected by BFA [105] provided considerable support to the present scheme proposing a role for the linker compartments in their intracellular trafficking (Figure 3). In fact, of the three alternative CoV release routes discussed above, the two previously presented ones are both expected to be sensitive to BFA (Figure 2), while the novel pathway introduced here (Figure 3) remains unaffected by this compound.

The study of Ghosh and coworkers further showed that, instead of following the Golgi-dependent constitutive secretory route to the extracellular space, β-CoVs employ the endo-lysosomal system for their cellular egress [105]. However, since the BFA-resistant route taken by the virus from the IC to the endosomal system was not further characterized, the unconventional pathway proposed in Figure 3 provides a viable option. Based on the employment of lysosomal membrane markers like LAMP-1, and identification of the exocytotic machinery involved, the authors arrived at the conclusion that the egress of CoVs takes place from lysosomes. Notably, however, the bulk of the analysis focuses on a late stage of CoV replication, when numerous changes have taken place in the infected cells. As discussed above, already at relatively early times of infection, due to luminal neutralization of secretory compartments, the Golgi apparatus undergoes disassembly, resulting in the redistribution of its residents. The authors’ demonstration that acidification of the endo-lysosomal system is also inhibited by CoVs makes it likely that the normal distribution of its components is affected, as well. Therefore, the possibility exists that while the endosomal recycling apparatus plays a key role in virus release at early times of infection, the cellular changes taking place over time allow CoVs to employ additional routes and exocytic mechanisms. 

Finally, a more general note concerning the use of viruses as models in cell biology. Since the mid-1970s, infection of cultured cells with various enveloped viruses was increasingly employed to study both the endocytic mechanisms operating during virus entry and the secretory apparatus that viruses exploit to gain exit from their host cells [11,148,149]. These studies also gave important insight into the overall pathways and mechanisms of intracellular membrane traffic, as well as cell polarity, particularly related to epithelial cells that often are at the frontline during a virus attack [47]. Moreover, viral mutants with temperature-sensitive defects in envelope protein transport provided useful tools for studies of the secretory process [150,151,152]. With the development of molecular biological methods in the 1980s, virus proteins and their mutated variants could be expressed and studied individually as marker cargo proteins for the intracellular transport routes of mammalian cells, which were beginning to display unexpected complexity. Furthermore, due to the subsequent development of microscopic techniques for live cell imaging, viral proteins were no longer obligatory gadgets in the toolbox of cell biologists, but still useful as markers for certain cellular compartments and pathways. In parallel, the detailed mechanisms operating during virus infection shifted away from the center stage of cell biology.

The ongoing COVID-19 pandemic, triggering an intensive search for pharmacological treatments to combat the disease—particularly, through repurposing of existing drugs—has been a clear reminder of the great importance of detailed knowledge on the different phases of the virus life cycle. To find out how a particular intracellularly budding virus harnesses the secretory mechanisms of its host cell for its own purposes one should focus on the early stages of infection—using as intact virus as possible—thus allowing the viral structural proteins to undergo unperturbed assembly in the host cell environment and acquire the post-translational modifications that they would naturally receive. However, to overcome the general effects of virus infection on the host cell—affecting over time the operation of membrane organelles and transport pathways—it can also be of interest to follow the assembly and release of VLPs or fluorescent virus particles by employing live cell imaging [108]. The results obtained thus far with CoVs and other intracellularly budding viruses should encourage future studies on the role of unconventional secretory pathways in their release. Related to viral pathogenesis, it would be of particular interest to investigate the role of these pathways in the polarized exit of viruses, such as CoVs [153,154], from different PM domains of epithelial cells.

## Figures and Tables

**Figure 1 cells-10-00503-f001:**
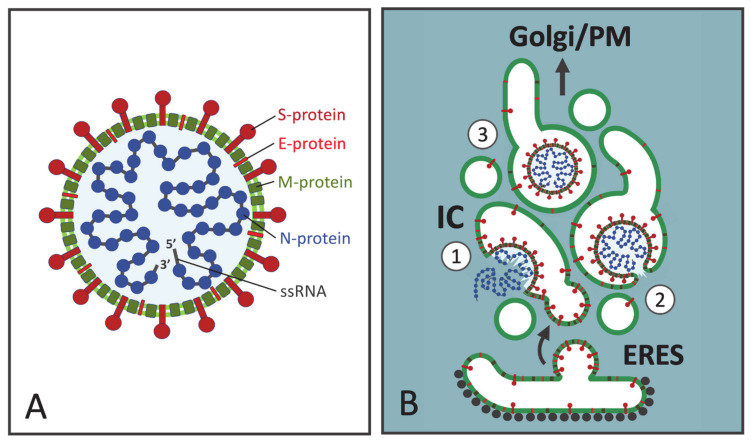
Structure and assembly of coronaviruses (CoVs). (**A**) The CoV virions consist of an outer envelope, a lipid bilayer (light green) supplemented by viral membrane proteins, and an inner helical nucleocapsid, consisting of multiple copies of the nucleocapsid protein (N, blue) bound to the single-stranded genomic RNA (ssRNA) of positive polarity. The spike protein (S, dark red) protruding from the envelope, gives the virion its characteristic corona-like appearance. The membrane single-spanning E protein (red) is present only in small quantities in the virus particles. The membrane multispanning M protein (dark green) is the most abundant membrane protein, which interacts with the other structural proteins and plays a key role in virus formation. (**B**) Virus assembly by budding at the intermediate compartment (IC) membranes located close to ER exit sites (ERES, See Appendix A). *Step 1*: N protein associates with newly made viral RNA genomes forming cytoplasmic nucleocapsids. Whether nucleocapsid assembly coincides with CoV budding, or preformed ribonucleoproteins are transported to the budding site is not known. The viral membrane proteins (S, E, and M) are synthesized on endoplasmic reticulum (ER)-associated ribosomes and packaged into transport vesicles for delivery to the IC. Virus budding is suggested to take place at vacuolar domains of the IC, which are large enough to accommodate the virus particles and may undergo further dilation. *Step 2*: Budding is probably largely based on specific interactions between the viral structural proteins. The process ends as the forming virus pinches-off from the membrane, due to membrane scission. *Step 3*: Following their entry into the IC lumen the virus particles are ready to move towards the plasma membrane (PM), as the saccular IC elements develop into mobile transport carriers. For simplicity, only one luminal CoV particle is shown, although the individual carriers may accommodate numerous progeny viruses.

**Figure 2 cells-10-00503-f002:**
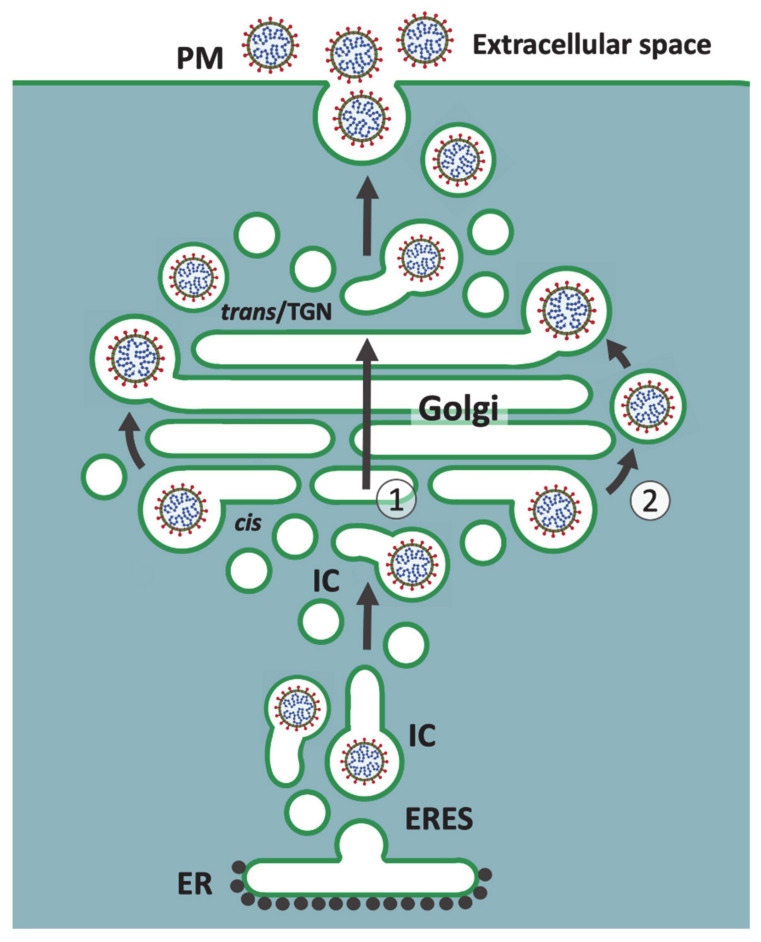
Two alternative current models of CoV release from host cells. Following their budding into the IC lumen, progeny CoVs are transported to the Golgi apparatus in IC-derived transport carriers. According to one scenario (***Route 1***) both the IC elements and Golgi *cisternae* are maturing structures. In this case, CoVs enter the Golgi stacks as the pre-Golgi carriers transform into *cis*-Golgi *cisternae* and are then transferred across the stacks following *cis*-to-*trans* movement of the maturing *cisternae*. Due to their large size, the virus particles are confined to the dilated ends of the mobile Golgi *cisternae*. Another possibility (***Route 2***) is that progeny CoVs enter “megavesicles” that bud from and fuse with the ends of Golgi *cisternae*, which in this case are considered as more static structures. Notably, both models propose that the transport of CoVs requires the integrity of the Golgi stacks. Following their arrival at *trans*-Golgi/TGN the virus particles are thought to be sorted to post-Golgi carriers for cell surface delivery along the constitutive secretory pathway. Of note, this illustration does not make a distinction between the pre- and post-Golgi forms of maturing CoV particles [16,89].

**Figure 3 cells-10-00503-f003:**
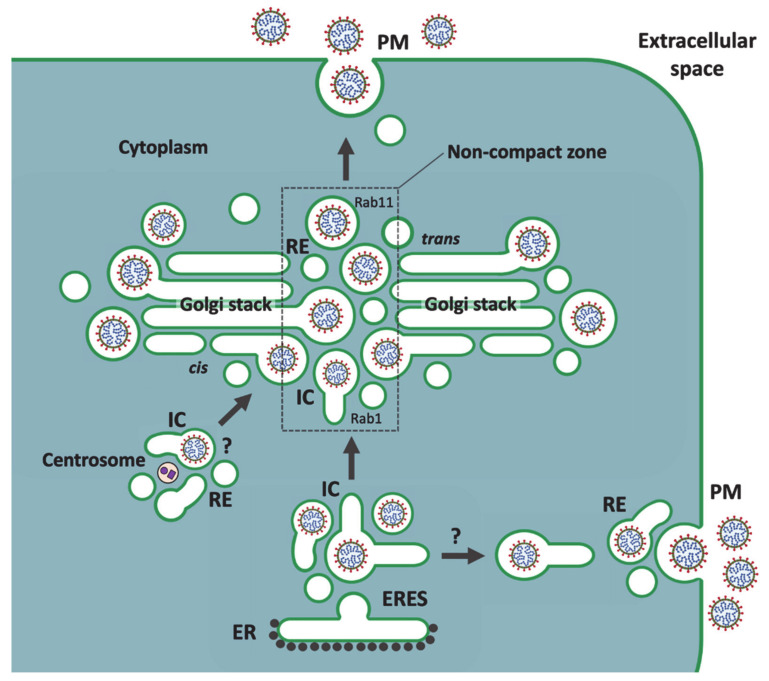
An unconventional secretory pathway that CoVs may utilize to reach the extracellular space. Following their budding into IC elements at ERES, CoVs reach the central Golgi region in mobile IC carriers. According to this model, the incoming pre-Golgi carriers, moving on MT tracks, preferentially arrive at the noncompact zones of the Golgi ribbon. Only two interconnected stacks of the ribbon are shown. Subsequently, the luminal virus particles are directly transferred from the Rab1-containing IC carriers to recycling endosomes (REs), thus entering the endocytic recycling circuit to the plasma membrane (PM), which is regulated by Rab11. The mechanism of this Golgi stack-independent transfer could involve a GTPase cascade that functionally links Rab1 and Rab11. At the noncompact zones, the saccular IC elements and REs also function in the biogenesis of the Golgi stacks, explaining the occasional presence of CoVs at the dilated rims of Golgi *cisternae*. The newly formed virus particles might also be transported directly from the ERES to the cell periphery [81], where communication of the IC carriers with REs leads to virus release. Whether CoV assembly also takes place at the IC elements that associate with the centrosome [82] remains currently unknown.

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
