# Peer review of "Assembly and Cellular Exit of Coronaviruses: Hijacking an Unconventional Secretory Pathway from the Pre-Golgi Intermediate Compartment via the Golgi Ribbon to the Extracellular Space"

_cells, 2021, doi:10.3390/cells10030503_

Round 1

Reviewer 1 Report

This is an extremely well-written and timely review article. The presentation is comprehensive, balanced and thoughtful. The authors provide the background required for an interested non-expert to understand the questions that the authors pose and their examination of the relevant evidence. The authors discuss several theories that are currently being offered to explain coronavirus budding, maturation and exit. While it is clear that they have a "favorite" theory that incorporates their views of the function of the IC in Golgi bypass pathways, they do not unduly emphasize this model but instead present it as one option in a broader context. The figures are extremely clear and nicely illustrate the complex concepts that are developed in the text. This review has the potential to be highly impactful both in the CoV field as for the broader cell biology community interested in the secretory pathway and the function of the Golgi complex. The only (very minor) issue that could benefit from some attention relates to the preponderance of abbreviations. A large number of abbreviations are used in the text and somewhat inconsistently defined. The non-expert reader would benefit from the addition of a glossary of abbreviations, or a more uniform policy of defining abbreviations, or both.

Author Response

Cover letter-1/Revised version of Saraste and Prydz (ID: cells-1102487)

Response to Reviewer-1:

We thank the reviewer for his/her positive response. Based on his/her suggestion, we have compiled a List of Abbreviations and appended it at the of the manuscript (before References).

We have also prepared a “Cell biology glossary” of the most relevant terms, as suggested by the reviewer, which hopefully could be included in the manuscript. It is provided as a separate file.

Reviewer 2 Report

The review by Saraste and Prydz is timely, thought-provoking and relevant to continued development of therapeutic interventions for deadly SARS-coronoaviruses. My recommendations for revisions are targeted to help the authors sharpen their central tenants for a broad audience that may not be fully versed in the sundry cell biology models of membrane trafficking.

Suggested reorganization to maintain focus and reader interest.

  1. The two sections at the outset of the review titled: The CoV budding compartment and The maturation strategies of CoVs are not unique (lines 51-126) feel bogged down with many old references and discussions of non CoVs at their start. I think that the authors would maintain greater audience interest if each of these sections started first with what is known about CoVs and then bringing in salient, up-to-date info on other viruses as relevant. E.g. sentence in lines 79-82 and 117-126 should be elevated to both launch and to conclude the discussions.
  2. Perhaps the section on CoV assembly at the IC membranes should be placed right up front and then weave in some supporting info related to other viruses beneath this section.
  3. The section title viral perspective on IC organization would benefit from clearly leveraging the information on ERGIC-53 dependence and BFA resistance of CoV transport. This would provide the framing for the most interesting arguments that come very late in the review re an unconventional, Golg-independent pathway of CoV egress via the intermediate compartment and recycling endosomes, analogous to large cellular cargo.

By cutting to the chase the expansive paragraphs long introduction to conventional Golgi transport (lines 225-287)  and vague references to alternative models could be minimized to one paragraph and moved further down into the body of the section, as it is unlikely to be the mechanism relevant to CoV egress as argued.

Specific comments:

  1. Lines 29-50. The introductory paragraph needs to start with a hook up front, e.g. Coronaviruses, like select other viruses, follow an unconventional biosynthetic route that entails budding into intracellular compartments instead of being assembled into budding competent structures at the plasma membrane…. And then go from there.
  2. Given focus of the review coronaviruses a bit more intro or a Table on the different genera and where CoV2 fits would be helpful for general audiences.
  3. While ERGIC-53 is introduced on line 121 significance is unclear until later. Add 2020 ref on HepB, which also exploits ERGIC-53 for egress--doi: 10.3390/cells9081889; and maybe introduce later.
  4. Adding an image of the pleomorphic CoV virions would be a nice addition as the reference to Fig. 1A does not support the statement. Perhaps add a comment how the pleomorphic sizes might come about based on the assembly and virus budding mechanisms? NB: Spelling pleomorphic instead of pleiomorphic.
  5. Line 144-150, add supporting references for the speculations and further explanation of how the E protein works in release of progeny viruses.
  6. Fig. 1B. It is impossible to distinguish the dark green M protein in panel B. Can you make this protein grey or black so that the viral proteins can be more easily seen in the budding sites?
  7. Line 176-177. Need to include and discuss newer published studies suggesting N protein is responsible for liquid-liquid phase separation to produce high local perinuclear concentrations of CoV nucleocapsids:
  • Nat Commun. 2020 Nov 27;11(1):6041. doi: 10.1038/s41467-020-19843-1.

  • Mol Cell. 2020 Dec 17;80(6):1078-1091.e6. doi:10.1016/j.molcel.2020.11.041. Epub 2020
  1. Lines 187-202. This paragraph is confusing due to jumping between different viruses in the discussions.
  2. Lines 203-223. The discussions of each of the viral protein functions is muddled and would benefit from greater clarity in the context of the trafficking Figure 1B.
  3. The model of IC to RE transport bypassing Golgi stacks, would be further supported by citations of Ladinsky and Howell showing ER association with both sides of Golgi stack. Also add Martínez-Martínez, N., Martínez-Alonso, E., Tomás, M., Neumüller, J., Pavelka, M., and Martínez-Menárguez, J. A. (2017).

Author Response

Cover letter-2/Revised version of Saraste and Prydz (ID: cells-1102487)

Response to Reviewer-2:

We thank the reviewer for his/her positive response and a number of constructive suggestions for improvement, most of which we have carefully considered and taken into account during the revision of the paper. These changes (indicated in the revised manuscript in red colour) are as follows:

Regarding general reorganization of the manuscript (comments 1-3 of the reviewer):

- Based on the reviewer’s suggestion – to maintain focus – we have revised the text in the two first sections of the manuscript; see: lines 51, 81-82, 87-91, 119-120. In addition, the title of the second section (line 86) has been changed to: “The assembly and exit strategies of CoVs are not unique”. However, we have maintained the overall organization of the three first sections. We think that the short historical sketch in the beginning on “the CoV budding compartment” is in place, as it also provides background information for the text that follows. The section “A viral perspective on IC organization” already contains relatively brief descriptions of the two previously proposed models on CoV release (related to Fig. 2), which in our opinion would not gain much from shortening into one paragraph. Also, the point that – unlike these two “old” models - our new model explains the BFA-resistance (independence of Golgi integrity) of CoV transport has been stated already in the end of the short introductory paragraph (lines 46-49).

Regarding specific comments (1-10) of the reviewer:

  1. Regarding the introductory paragraph, we think that the “hook” suggested by the reviewer is already contained in the title of paper, as well as in the Abstract.
  2. To highlight the fact that SARS- and MERS CoVs belong to the beta-coronaviruses, the text in lines 82-85 has been revised.
  3. The paper on the role of ERGIC-53 in hepatitis B virus replication, pointed out by the reviewer, has been referred to – see line 91, as well as included to the reference list (#150). We have also added another reference (#149), suggesting a role for the IC in the life cycle of hepatitis B virus.
  4. Since the pleomorphic structure of CoVs is not a central issue here, we have not included a negatively stained EM image of the viruses. Also, Fig. 1B indicates the size variation of the virus. However, we have added a comment on the pleomorphic nature of CoV virions in lines 214-216. We now consistently use “pleomorphic”, instead of “pleiomorphic”.
  5. The requested supporting references (#4 and #46) have been included (line 149). Also, the text regarding the role of the E protein in CoV release, largely based on studies of IBV, has been revised. See: lines 220-221, 245, 404.
  6. To highlight the M protein better at the budding sites, Fig. 1B has been modified. The revised Figure 1 is included as a separate file, and has not yet been incorporated into the manuscript.
  7. The new interesting data on the possible role of liquid-liquid phase separation in CoV assembly has been mentioned in lines 177-179. The two references suggested by the reviewer (#151 and #152) have been added to the reference list.
  8. The particular paragraph (lines 191-206) has been revised (lines 193, 196) to better distinquish between the different CoVs used in the various studies.
  9. With the above mentioned changes incorporated in these two paragraphs (lines 207-229), we hope that the text is clearer now.
  10. One of the papers suggested by the reviewer (Ladinsky et al., 1999) has been referred to (line 333) and added to the reference list (#154).

Reviewer 3 Report

The authors describe possible mechanisms of Coronavirus assembly and budding from the cell host. The article is nicely written, well organized and informative. I do not have any specific comments. 

Author Response

Cover letter-3/Revised version of Saraste and Prydz (ID: cells-1102487)

Response to Reviewer-3:

We thank the reviewer for his/her positive response.